

# Freshwater reservoir offsets on radiocarbon-dated dog bone from the headwaters of the St. Lawrence River, USA

John P. Hart[1], Robert S. Feranec[1], Timothy J. Abel[2] and Jessica L. Vavrasek[1,3]

[1] Research and Collections Division, New York State Museum, Albany, NY, United States of America
[2] Carthage, NY, United States of America
[3] Department of Anthropology, University at Albany, Albany, NY, United States of America

## ABSTRACT

Isotopic analysis of dog (*Canis lupus familiaris*) bone recovered from archaeological sites as proxies for human bone is becoming common in North America. Chronological placement of the dogs is often determined through radiocarbon dating of dog bone. The Great Lakes, their tributaries, and nearby lakes and streams were important fisheries for Native Americans prior to and after sustained European presence in the region. Carbon entering the food web in freshwater systems is often not in full isotopic equilibrium with the atmosphere, giving rise to spuriously old radiocarbon ages in fish, other aquatic organisms, and their consumers. These freshwater reservoir offsets (FROs) have been noted on human and dog bone in several areas of the world. Here we report the results of multi-tracer Bayesian dietary modeling using $\delta^{15}N$ and $\delta^{13}C$ values on dog bone collagen from mid-fifteenth to mid-sixteenth-century Iroquoian village sites at the headwaters of the St. Lawrence River, New York, USA. Results indicate that fish was an important component of dog diets. A comparison of radiocarbon dates on dog bone with dates on deer bone or maize from the same sites indicate FROs ranging from $97 \pm 24$ to $220 \pm 39$ $^{14}$Cyr with a weighted mean of $132 \pm 8$ $^{14}$Cyr. These results suggest that dog bone should not be used for radiocarbon dating in the absence of modeling to determine fish consumption and that previously reported radiocarbon dates on human bone from the larger region are likely to have FROs given the known importance of fish in regional human diets.

Corresponding author
John P. Hart, john.hart@nysed.gov

## INTRODUCTION

The canine surrogacy approach is becoming increasingly popular in North American archaeology to assess human diets through isotope analyses when Native American human bone is unavailable for destructive analysis. The assumption is that dogs (*Canis lupus familiaris*) had diets similar to those of the humans with whom they lived (*Edwards, Jeske & Coltrain, 2017*). These analyses of dog bone may also include radiocarbon assays to determine the chronological placements of the dog remains under study or more broadly

of features and layers within the particular sites where the dog bones were found. Dog bone may also be radiocarbon dated when used for other analyses such as ancient DNA (e.g., *Perri et al., 2019*).

The North American Great Lakes region, including the lakes, their tributaries, nearby smaller lakes and their tributaries, and the St. Lawrence River were important Native American fisheries (*Cleland, 1982*). The zooarchaeological record attests to the importance of freshwater fish in regional subsistence systems prior to sustained European presence in the region (e.g., *Hawkins et al., 2019*). Ethnohistoric records indicate the importance of freshwater fish in both human (*Heidenreich, 1971*) and dog (*Lovis & Hart, 2015*) diets in this and surrounding areas during the seventeenth-century AD and after.

Freshwater bodies may harbor ancient carbon, eroded from bedrock and unconsolidated sediments and soils, which is metabolized and incorporated into fish tissues and the tissues of fish consumers (e.g., *Keaveney & Reimer, 2012*; *Hart, Taché & Lovis, 2018*). As a result, radiocarbon ages on the bones of fish and fish consumers may be significantly older than the actual ages of the bones. Such freshwater reservoir offsets (FRO) have been documented on human bone in regions where freshwater fish were consumed by comparing dates on human bone to dates on the bone of terrestrial herbivores and plant remains from a given archaeological site or site component—so-called contextual dates (e.g., *Schulting et al., 2014*; *Lillie et al., 2016*). FROs may also be present in radiocarbon dates on charred cooking residues adhering to pottery that incorporate carbon from fish (*Heron & Craig, 2015*).

Given that dogs in ethnohistorically recorded seventeenth-century AD lower Great Lakes Iroquoian villages and towns were scavengers and were also fed leftover food prepared for human consumption (*Heidenreich, 1971*), and that there is ample evidence for human consumption of freshwater fish in the ethnohistorical record and at Iroquoian archaeological sites dating to earlier centuries, it is likely that freshwater fish were components of dog diets. Here we examine stable carbon ($\delta^{13}$C) and nitrogen ($\delta^{15}$N) isotope values using Bayesian dietary mixing models to test the hypothesis that dog diets at mid-fifteenth to mid-sixteenth-century AD Iroquoian villages at the headwaters of the St. Lawrence River included substantial amounts of fish.

There are 65 documented Iroquoian archaeological village and related sites south of the headwaters of the St. Lawrence River and east of Lake Ontario in present-day northern New York (Fig. 1). A recent program of radiocarbon dating maize (*Zea mays* ssp. *mays*) kernels and white-tailed deer (*Odocoileus virginianus*) bone from 18 of these sites and Bayesian modeling of the resulting 43 $^{14}$C ages suggests that the total occupation span was less than 100 years, from the mid fifteenth-century AD to the mid-sixteenth-century AD (*Abel, Vavrasek & Hart, in press*). Given the large number of village components it is likely that each village was occupied for a short period of time and not reoccupied. Changes in pollen and charcoal records from an area lake core demonstrate anthropogenic impacts to the region from land clearance during this time span. This was followed by a period of recovery after abandonment of the area by Iroquoians and before the onset of historical land clearance beginning in the nineteenth century AD (*Brown, 2002*). There is little evidence for human occupations of the area immediately prior to the Iroquoian occupations, the last well-defined occupations occurring several hundred years earlier (*Abel*
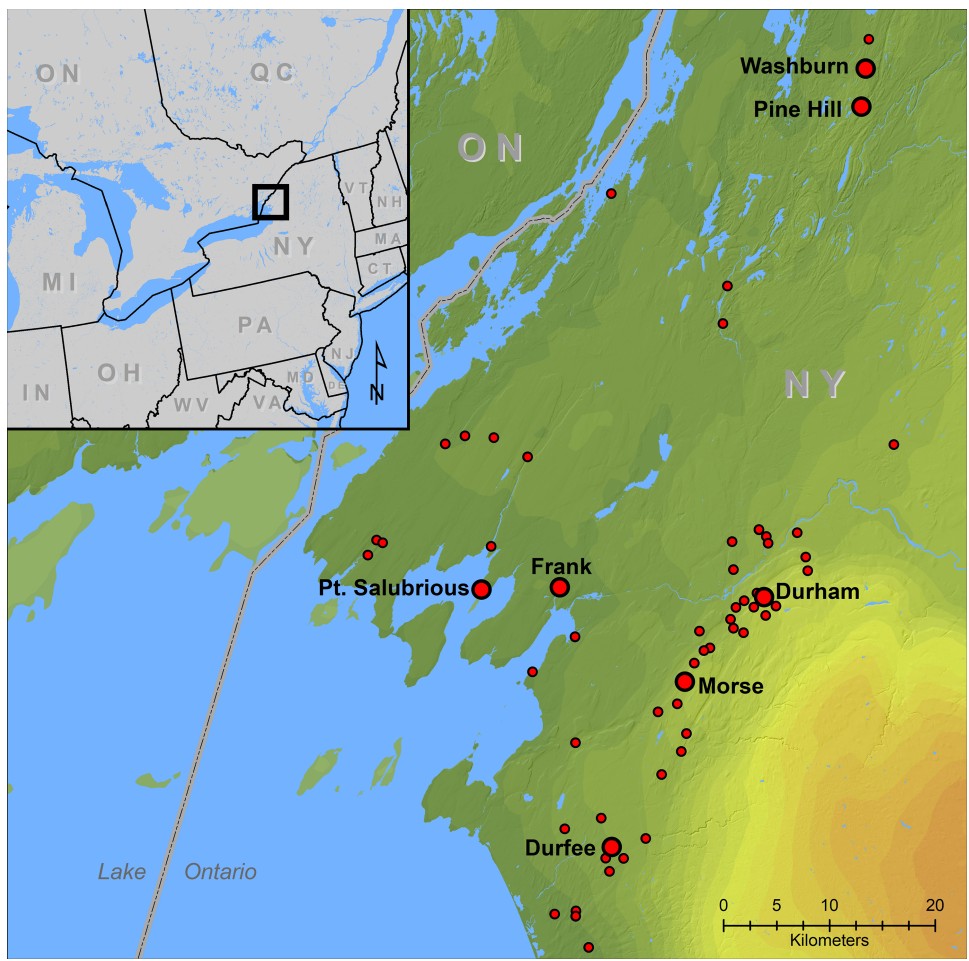

**Figure 1** **Locations of Iroquoian sites at the headwaters of the St. Lawrence River USA.** Large labeled dots are sites with radiocarbon dated dog bone.

*& Fuerst, 1999*). These various lines of evidence suggest that materials recovered from each of the Iroquoian sites relate to a one- or two-decade occupation.

After modeling dog diets, we next determine if radiocarbon dates on dog bones result in FROs. We do this by comparing [14]C ages on dog bone collagen with contextual [14]C ages on white-tailed deer bone collagen or maize kernels from the same sites. We then discuss the likelihood that radiocarbon dates on human tissue (collagen) will have significant FROs.

## MATERIALS & METHODS

Samples of dog and deer bone included in this analysis are from collections obtained by the New York State Museum in the early twentieth century. With the exception of the St. Lawrence site samples, which are identified as all having come from the same pit feature by the excavator, these collections generally lack other than site-level provenance data. However, the samples used from each site were obtained from a specific collector increasing confidence that they came from the same site deposits. There is no evidence

for use of consolidants on any of the bone. For Durfee and Pine Hill the dates available for comparison with those on dog bone were on maize kernels. For Durfee there is only site-level provenance data, while for Pine Hill, both maize samples are documented as having come from the same pit feature.

Dog and deer bone fragments were submitted to the W. M. Keck Carbon Cycle Accelerator Mass Spectrometry Laboratory of the University of California, Irvine (KCCAMS). These were first decalcified with 0.5N HCl, then gelatinized at 60 °C and pH 2, and finally ultrafiltered to obtain molecular weight fractions >30 kDa (*Beaumont et al., 2010*). Ultrafiltered collagen aliquots were measured to a precision of <0.2‰ for $\delta^{15}N$ (traceable to AIR) and <0.1‰ for $\delta^{13}C$ (traceable to PDB). Maize samples were submitted to KCCAMS or the University of Georgia Center for Applied Isotope Studies (CAIS) where they received standard acid–base-acid pretreatments prior to combustion. All maize $\delta^{13}C$ values were measured to a precision of <0.1‰ relative to standards traceable to PDB both facilities on pretreated aliquots. Maize samples were measured to a precision of <0.2‰ for $\delta^{15}N$ (traceable to AIR) at KCCAMS. AMS dates were corrected for isotopic fractionation and reported according to established conventions (*Stuiver & Polach, 1977*). Each facility's website can be visited for additional information on their respective protocols (https://sites.uci.edu/keckams/protocols/; http://cais.uga.edu/).

Bayesian two tracer dietary mixing models were run in MixSIAR v 3.1.10 (*Stock et al., 2018*; *Stock & Semmens, 2016*) using $\delta^{13}C$ and $\delta^{15}N$ values on bone and maize. MixSIAR uses a Markov Chain Monte Carlo (MCMC) simulation to model the proportions of sources in a consumer's diet based on the stable isotope values of the consumer and its food. Additionally, MixSIAR incorporates the uncertainty in the trophic enrichment factor between the food sources and the consumer in the model. Data for prey (source) animal species were obtained from *Booth (2014)*, *Guiry et al. (2016)*, *Morris (2015)*, *Morris et al. (2016)* and *Pfeiffer et al. (2016)*. Bone reported from these sources span approximately AD 900–1,600 with 6 of 44 samples reported in *Guiry et al. (2016)* dating to the historical era. Although adequate samples of terrestrial and aquatic prey species are not available from northern New York, statistical tests (see below) comparing dog and deer data from northern New York to that from southern Ontario show no statistically significant differences supporting their use for the source groups. Additionally, the fish data derive from the same main watersheds, Lake Ontario and the St. Lawrence River, that would have been utilized by peoples in northern New York. Assayed fish bone is from archaeological sites bordering the north shore of Lake Ontario and its tributaries in southern Ontario and the St. Lawrence River in Quebec, with the exception 7 of 91 samples reported by *Pfeiffer et al. (2016)* that are from more northerly southern Ontario sites. Given that the northern New York sites border Lake Ontario and the St. Lawrence River, the isotope values reported for the Ontario sites are reasonable proxies for the current analyses.

The diversity of potential food sources was placed into seven sources in the mixing models. Additional models using the same seven sources as well as models using only three sources were performed for individual dogs (Table S1). The seven-source model was our preferred model as it provided the fewest total sources while taking into account taxonomy, ecology, differences in body size (e.g., small, large), and statistically significant

differences in stable isotope values. For example, fish isotope values were first compared based on their ecological groupings as in *Pfeiffer et al. (2016)*, and ultimately split into three separate sources based on statistically significant differences among taxa in $\delta^{15}N$ values. Black bear and white-tailed deer were combined into one source as there were no statistically significant differences between those species in either $\delta^{15}N$ or $\delta^{13}C$ values. Sources for the seven source models include: (1) maize, (2) high $\delta^{15}N$ fish, (3) medium $\delta^{15}N$ fish, (4) low $\delta^{15}N$ fish, (5) bear (*Ursus* spp.) and deer (*Odocoileus virginianus*), (6) small herbivorous mammals, and (7) turkey (*Meleagris gallopavo*), while the sources in the three source models were (1) maize, (2) deer, and (3) all fish (Supplemental Data S1). Stable isotope values for $C_3$ plants were not included in the models as we have no isotopic data for archaeological $C_3$ plants and using modern data would likely fall among and not be distinguishable from the "small herbivorous mammals" group. Collagen to collagen source (prey) to consumer (dogs) trophic enrichment factors (TEF) for $\delta^{13}C$ (+1.1‰ ± 0.2‰) and $\delta^{15}N$ (+3.8‰ ± 1.1‰) were derived from *Bocherens et al. (2015)*, which is calculated from an average of studies on different taxa as there is not a specific TEF calculated for these archaeological dogs. The maize (source) to consumer (dogs) TEF was +5.0‰ ± 0.1‰ for $\delta^{13}C$, and +3.0‰ ± 0.1‰ for $\delta^{15}N$ (*Ambrose et al., 1997*). All Bayesian mixing models met criteria of the Gelman–Rubin and Geweke tests and are reported in percent credible intervals (posterior probabilities).

All radiocarbon date calibrations, tests for significant differences between dates, and Bayesian modeling of radiocarbon dates were done in OxCal 4.3 (*Bronk Ramsey, 2009a*; *Bronk Ramsey, 2009b*). Calibrations and modeling used the IntCal13 Northern Hemisphere terrestrial calibration dataset (*Reimer et al., 2013*). No artifacts manufactured from European materials have been found on any of the sites in question relating to the Iroquoian occupations, which indicates the sites were occupied prior to widespread adoption of European materials in the late sixteenth and early seventeenth-century AD (*Loewen & Chapdelaine, 2016*; *Manning et al., 2018*). However, because there is a plateau in the calibration curve affecting calibrations in the sixteenth-century AD, modeled dates are multimodal and may extend well into the seventeenth century AD. To correct for this problem, a *terminus ante quem* (*TAQ*) of AD 1,600 ± 10 was used in modeling. The CQL run file for each model is presented in Supplemental Code S1.

Following *Keaveney & Reimer* (*2012*:1210), FROs were calculated by subtracting uncalibrated dog bone $^{14}C$ ages from those on deer bone or maize, and FRO standard deviations were calculated through propagation of errors ($\sqrt{\sigma^2_{dog} + \sigma^2_{context}}$). In cases where there were multiple dates on either deer bone or maize from a given site, they were combined with the OxCal R_Combine();in each case the dates were not significantly different at the 95% level of confidence, and their weighted mean was used in FRO calculations (*Ward & Wilson, 1978*).

## RESULTS

$\delta^{13}C$ and $\delta^{15}N$ values for dogs in our sample, dogs from southern Ontario sites, and seven-source model source groups are presented in Fig. 2. As is evident the isotope values for dogs

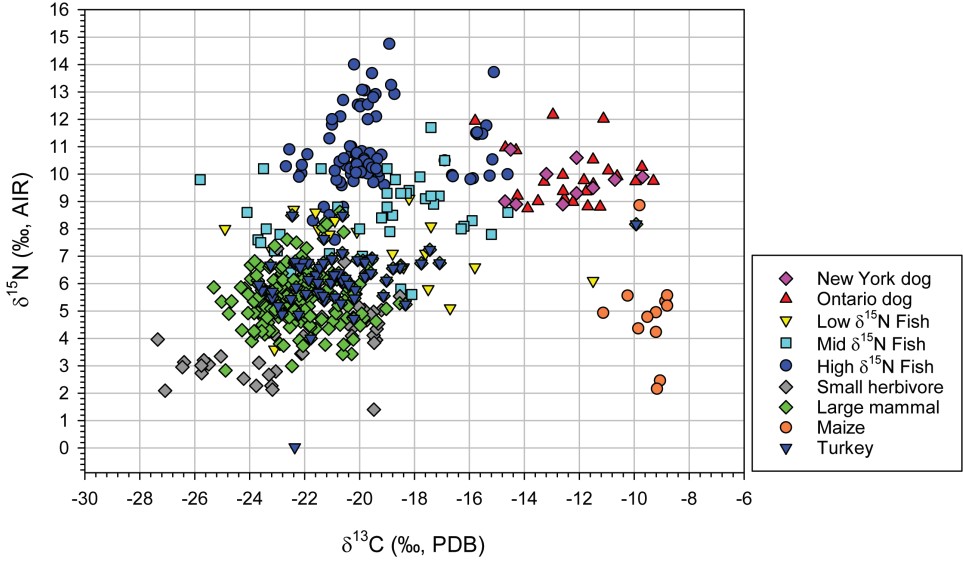

**Figure 2** Scatter plot of $\delta^{13}$C and $\delta^{15}$N values for dogs and prey groups used in the seven source Bayesian dietary mixing model.

from New York plot among those for dogs from southern Ontario. Mann–Whitney U tests indicate no statistical difference between New York ($n = 10$) and southern Ontario ($n = 24$) dog isotope values: permutation $p = 0.544$ for $\delta^{13}$C and 0.681 for $\delta^{15}$N. Mann–Whitney U tests also indicate no statistical difference between New York ($n = 8$) and a random sample of southern Ontario deer values ($n = 20$): permutation $p = 0.478$ for $\delta^{13}$C and 0.892 for $\delta^{15}$N. These results suggest use of isotopes from southern Ontario in the Bayesian mixing models for potential prey sources is warranted.

Results of the Bayesian dietary mixing modeling for the dogs as a group are presented in Table 1. The sum of mean values for fish for the seven-source model is 0.250, while maize is 0.475, and the sum of remaining terrestrial resource means is 0.275. Dietary mixing modeling results for individual dogs are presented in Table S1. Sums of mean values for fish in the seven-source models for individual dogs range from 0.228 to 0.444. Given these results, it is evident that we can reject the null hypothesis that freshwater fish were not an important component of dog diets.

Results of radiocarbon assays on dog and contextual deer bone and maize are presented in Table 2 by site. C/N ratios for all bone samples are within the acceptable range for radiocarbon dating and isotopic assay (*Ambrose et al., 1997*; *Beaumont et al., 2010*; *Van Klinken, 1999*). The $^{14}$C ages on dog bone collagen are consistently older than those on the context materials (Table 3). Tests of sample significance (*Ward & Wilson, 1978*) between $^{14}$C ages on dog bone and $^{14}$C ages on deer bone or maize were performed with the R_Combine (); command in OxCal. In each case the $T$-value exceeds the 0.05 $\chi^2$ critical value (3.84) for one-degree of freedom (Table 3), indicating the dog $^{14}$C ages are significantly older than those of deer bone or maize. Calculations of offsets for $^{14}$C ages on dog bone compared to dates on deer bone or maize are presented in Table 3. The offsets

**Table 1  Bayesian dietary mixing model results for all dogs as a group.**

|  | Mean | SD | 2.5% | 5% | 25% | 50% | 75% | 95% | 97.5% |
|---|---|---|---|---|---|---|---|---|---|
| **Table 1A** | | | | | | | | | |
| Seven source model | | | | | | | | | |
| Bear and deer | 0.097 | 0.069 | 0.005 | 0.009 | 0.040 | 0.084 | 0.142 | 0.227 | 0.254 |
| Maize | 0.475 | 0.025 | 0.425 | 0.433 | 0.459 | 0.475 | 0.492 | 0.513 | 0.521 |
| High $\delta^{15}$N fish | 0.076 | 0.051 | 0.004 | 0.007 | 0.035 | 0.070 | 0.110 | 0.170 | 0.192 |
| Low $\delta^{15}$N fish | 0.088 | 0.072 | 0.001 | 0.003 | 0.031 | 0.071 | 0.130 | 0.222 | 0.264 |
| Medium $\delta^{15}$N fish | 0.086 | 0.063 | 0.005 | 0.009 | 0.034 | 0.073 | 0.123 | 0.208 | 0.239 |
| Small mammal herbivores | 0.081 | 0.055 | 0.004 | 0.007 | 0.036 | 0.073 | 0.117 | 0.184 | 0.203 |
| Turkey | 0.098 | 0.075 | 0.005 | 0.008 | 0.037 | 0.082 | 0.143 | 0.242 | 0.275 |
| **Table 1B** | | | | | | | | | |
| Three-source model | | | | | | | | | |
| Maize | 0.481 | 0.040 | 0.400 | 0.416 | 0.456 | 0.480 | 0.506 | 0.545 | 0.532 |
| Deer | 0.314 | 0.078 | 0.160 | 0.180 | 0.263 | 0.316 | 0.366 | 0.441 | 0.464 |
| Fish | 0.205 | 0.075 | 0.058 | 0.084 | 0.154 | 0.204 | 0.256 | 0.331 | 0.355 |

range from 97 ± 24 to 220 ± 39 $^{14}$Cyr with a weighted mean of 132 ± 8. *Schulting et al. (2014)* found a significant linear relationship between $\delta^{15}$N values and offsets between human bone and FROs in human bone $^{14}$C ages from Lake Baikal, Siberia. A regression of $\delta^{15}$N values on FRO means for our sample resulted in a moderate positive correlation that is not significant (permutation $p = 0.180$). This may be a result of small sample size or varying dog diets. Given the short occupation span of this geographically restricted area, it seems less likely, but is possible, that it results from localized temporal or spatial differences in fish $\delta^{15}$N values or carbon reservoirs.

## DISCUSSION

Freshwater fish was an important component of prehistoric human diets in many areas of the world as is evident from the recovery of fish bone on archaeological sites (e.g., *Hawkins et al., 2019*), analysis of lipids recovered from pottery (e.g., *Craig et al., 2013*), and isotopic analysis of human bone (e.g., *Lillie et al., 2016*). Analyses of dog bone from prehistoric archaeological sites has demonstrated that fish was also an important component of some dog diets (e.g., *Fischer et al., 2007*). Given the widely recognized phenomenon of the freshwater reservoir effect on human bone as well as the bone of dogs that were fed fish (e.g., *Losey et al., 2018*), it is important to consider the potential for FROs on dog bone by assessing the likelihood that dogs on a given site were fed fish in areas with freshwater bodies having the potential for ancient carbon reservoirs in the past.

Here we have demonstrated that dogs at mid-fifteenth to mid-sixteenth-century Iroquoian sites at the headwaters of the St. Lawrence River had diets that included freshwater fish and that radiocarbon dates on dog bone have offsets relative to terrestrial sources. Because the dog diets included substantial amounts of fish that contributed to collagen formation, these offsets are evidently FROs. While the offsets on dog bone are modest compared to those in some areas (e.g., *Losey et al., 2018*), they are still significant

**Table 2  Radiocarbon dates used in analyses.**

| Lab # | Site | Material | $\delta^{13}$C (‰) | $\delta^{15}$N (‰) | Collagen C/N$_{atomic}$ | $^{14}$C age (BP) |
|---|---|---|---|---|---|---|
| UCIAMS-199804 | Durfee | dog bone | −9.7 | 9.9 | 3.26 | 460 ± 15 |
| UCIAMS-205978 | Durfee | maize | −9.0 | | | 345 ± 15 |
| UGAMS-31486 | Durfee | maize | −10.1 | | | 336 ± 22 |
| UGAMS-34185 | Durfee | maize | −10.2 | | | 314 ± 22 |
| UCIAMS-199798 | Frank | dog bone | −14.3 | 8.9 | 3.28 | 535 ± 20 |
| UCIAMS-199801 | Frank | dog bone | −13.2 | 10.0 | 3.25 | 580 ± 15 |
| UCIAMS-199802 | Frank | deer bone | −22.9 | 3.8 | 3.23 | 410 ± 15 |
| UCIAMS-199803 | Morse | dog bone | −12.1 | 9.3 | 3.42 | 510 ± 20 |
| UCIAMS-199806 | Morse | deer bone | −22.3 | 7.5 | 3.41 | 410 ± 15 |
| UCIAMS-204718 | Morse | deer bone | −21.0 | 6.4 | 3.33 | 420 ± 25 |
| UCIAMS-204721 | Pine Hill | dog bone | −12.6 | 8.9 | 3.21 | 550 ± 25 |
| UCIAMS-199800 | Pine Hill | dog bone | −10.7 | 9.8 | 3.23 | 490 ± 15 |
| UGAMS-37380 | Pine Hill | maize | −9.2 | | | 353 ± 20 |
| UGAMS-37381 | Pine Hill | maize | −9.0 | | | 392 ± 20 |
| UCIAMS-199807 | Point Salubrious | dog bone | −14.7 | 9.0 | 3.33 | 515 ± 15 |
| UCIAMS-199805 | Point Salubrious | deer bone | −22.8 | 5.7 | 3.22 | 405 ± 15 |
| UCIAMS-204715 | St Lawrence | dog bone | −12.1 | 10.6 | 3.22 | 610 ± 25 |
| UCIAMS-204717 | St Lawrence | dog bone | −14.5 | 10.9 | 3.23 | 705 ± 30 |
| UCIAMS-204714 | St Lawrence | deer bone | −22.9 | 5.7 | 3.23 | 485 ± 25 |
| UCIAMS-199799 | Washburn | dog bone | −11.5 | 9.5 | 3.28 | 525 ± 15 |
| UCIAMS-204722 | Washburn | deer bone | −22.3 | 6.2 | 3.16 | 415 ± 30 |

**Table 3  Calculation of freshwater reservoir offsets (FRO) for dog bone dates.**

| Site | Context material | Dog bone UCIAMS # | Context date Lab# | $^{14}$C$_{Dog}$ | $^{14}$C$_{Context}$ | $\chi^2$ T[a] | FRO |
|---|---|---|---|---|---|---|---|
| Durfee | Maize pooled mean | 199804 | UCIAMS-205978 UGAMS-31485 UGAMS-31486 | 460 ± 15 | 335 ± 11 | 45.4 | 125 ± 18 |
| Frank | Deer bone | 199798 | UCIAMS-199802 | 535 ± 20 | 410 ± 15 | 25.1 | 125 ± 25 |
| Frank | Deer bone | 199801 | UCIAMS-199802 | 580 ± 15 | 410 ± 15 | 64.2 | 170 ± 21 |
| Morse | Deer bone pooled mean | 199803 | UCIAMS-199806 UCIAMS-204718 | 510 ± 20 | 413 ± 13 | 16.6 | 97 ± 24 |
| Pine Hill | Maize pooled mean | 204721 | UGAMS-37380 UGAMS-37381 | 550 ± 25 | 373 ± 15 | 37.2 | 177 ± 29 |
| Pine Hill | Maize pooled mean | 199800 | UGAMS-37380 UGAMS-37381 | 490 ± 15 | 373 ± 15 | 30.4 | 117 ± 21 |
| Point Salubrious | Deer bone | 199807 | UCIAMS-199805 | 515 ± 15 | 405 ± 15 | 29.6 | 110 ± 21 |
| St. Lawrence | Deer bone | 204715 | UCIAMS-204714 | 610 ± 25 | 485 ± 25 | 12.5 | 125 ± 35 |
| St. Lawrence | Deer bone | 204717 | UCIAMS-204714 | 705 ± 30 | 485 ± 25 | 31.9 | 220 ± 39 |
| Washburn | Deer bone | 199799 | UCIAMS-204722 | 525 ± 15 | 415 ± 30 | 10.7 | 110 ± 34 |

**Notes.**
[a]$df = 1$, 0.05 $\chi^2$ critical point = 3.84.

given that dates on deer and maize suggest a short total occupation span for the area, beginning in the fifteenth century AD. If assumed to be accurate the dates on dog bone would extend the Bayesian-modeled occupation span into the fourteenth century AD (Table 4). Our results combined with those of other recent assessments of radiocarbon

**Table 4  Radiocarbon Bayesian modeling results at 95.4% probability.**

| Model | Start boundary | Undated event | End boundary | $A_{model}$ | $A_{overall}$ |
|---|---|---|---|---|---|
| Deer and maize dates | 1421–1440 | 1431–1546 | 1513–1590 | 82.5 | 75.7 |
| Deer, dog, and maize dates | 1364–1396 | 1378–1563 | 1521–1606 | 78.6 | 76.2 |

dates on dog bone (e.g., *Losey et al., 2018*) indicate analysts should assess the potential for FROs prior to radiocarbon-dating dog bone.

While our sample is small and limited to the St. Lawrence River headwaters, the $\delta^{13}C$ and $\delta^{15}N$ values are not significantly different from those obtained on contemporaneous dog bone from southern Ontario, which suggests similar levels of fish consumption. Published chronological assessments of several Iroquoian ossuaries in southern Ontario are based on one or a few radiocarbon dates on human tissue (e.g., *Williamson & Pfeiffer, 2003*; *Pfeiffer et al., 2017*). Given evidence for the importance of fish in human diets from that area, there is a possibility of FROs in radiocarbon dates on human bone and dentine collagen. For example, a $^{14}C$ age on maize from the Moatfield Ossuary of $620 \pm 60$ BP contrasts with $^{14}C$ ages on human bone of $730 \pm 40$ BP, $810 \pm 40$ BP, and $910 \pm 40$ BP (*Williamson, Thomas & MacDonald, 2003*:82). While the older ages on human tissue may have other explanations, FROs cannot be discounted based on current data; freshwater fish were an important component of the zooarchaeological record at the associated Moatfield village site (*Williamson, Thomas & MacDonald, 2003*:53–73). The dates on human bone from Moatfield and other Iroquoian ossuaries should be reevaluated in light of our results, as should any radiocarbon dates on dog bone.

## CONCLUSION

Freshwater reservoir offsets in radiocarbon dates on source material incorporating carbon from aquatic resources is a global phenomenon. Research to date in archaeology has focused primarily on fish, human bone, and charred cooking residues adhering to pottery. Less attention has been paid to the bone of other animals who may have consumed fish. Here we have demonstrated that offsets are evident in $^{14}C$ ages on the bone of 10 dogs from mid-fifteenth- to mid-sixteenth-century AD Iroquoian village sites at the headwaters of the St. Lawrence River, USA. Bayesian dietary mixing models indicate that fish were important components of the dogs' diets. It is likely, then, that the offsets in the $^{14}C$ ages on dog bone relative to $^{14}C$ ages on deer bone or maize kernels, are FROs. Archaeologists pursuing the canine surrogacy approach to assess human diets in North America and elsewhere through isotopic analysis should take the possibility of FROs into account prior to obtaining radiocarbon dates on the analyzed dog bone. Our results also suggest the need to reevaluate radiocarbon dates obtained on human bone in nearby southern Ontario, where the archaeological and ethnohistoric records indicate fish were important components of Iroquoian diets.

## ACKNOWLEDGEMENTS

We thank Susan Winchell-Sweeney for producing Fig. 1. We thank the two peer reviewers for their well-considered comments and suggestions.

### Funding

The authors received no external funding for this work. The funders had no role in study design, data collection and analysis, decision to publish, or preparation of the manuscript.

### Competing Interests

The authors declare there are no competing interests.

### Author Contributions

- John P. Hart conceived and designed the experiments, performed the experiments, analyzed the data, prepared figures and/or tables, authored or reviewed drafts of the paper, approved the final draft.
- Robert S. Feranec performed the experiments, analyzed the data, prepared figures and/or tables, authored or reviewed drafts of the paper, approved the final draft.
- Timothy J. Abel and Jessica L. Vavrasek contributed reagents/materials/analysis tools, authored or reviewed drafts of the paper, approved the final draft, edited the draft paper.

### Data Availability

The raw data used with the program MixSIAR are available in Data S1. The CQL code used with OxCal 4.3 is available in Code S1.

### Supplemental Information

Supplemental information for this article can be found online at http://dx.doi.org/10.7717/peerj.7174#supplemental-information.

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
