# Peer review of "Freshwater reservoir offsets on radiocarbon-dated dog bone from the headwaters of the St. Lawrence River, USA"

_PeerJ, doi:10.7717/peerj.7174_

## Round 0.1 · original submission · Minor Revisions

The manuscript is interesting, but as pointed by the reviewers, it needs few changes. In the rebuttal letter, please respond to all comments of the reviewers.

# ·

Basic reporting

I have made a small number of suggestions for improving the clarity of the English.

Experimental design

I have noted a couple of small mistakes in formulae used, which do not however affect the interpretation.

Validity of the findings

Compared to other research in this field, including my own, the data set seems pretty robust. I have raised a couple of questions about the applicability of the published data used to interpret the new results - which I think the authors will be able to address.

Additional comments

please see the uploaded PDF version of my comments

·

Basic reporting

Language: no comments.

References/background:
In the Introduction (see lines 42-43), the discussion about the suitability of dog diet as proxy for human diet could be expanded. I attach a few references with arguments for (Noe-Nygaard 1988, Clutton-Brock and Noe-Nygaard 1990) and against (Eriksson & Zagorska 2003; Fischer et al. 2007: Fig. 5B and 5C) this approach. Some of these were already referred to in Edwards et al. 2017, which you cite in the Introduction.

Noe-Nygaard N. 1988. 13C-values of dog bones reveal the nature of changes in man's food resources at the mesolithic-neolithic transition, Denmark. Chemical Geology: Isotope Geoscience section 73(1):87-96.
Eriksson G, Zagorska I. 2003. Do dogs eat like humans? Marine stable isotope signals in dog teeth from inland Zvejnieki. In: Larsson L, Kindgren H, Knuttson K, Loefler D, Åkerlund A, editors. Mesolithic on the Move. Papers presented at the Sixth International Conference on the Mesolithic in Europe. Oxford: Oxbow Books. p 160-8.
Fischer A, Olsen J, Richards M, Heinemeier J, Sveinbjörnsdottir ÁE, Bennike P. 2007. Coast-inland mobility and diet in the Danish Mesolithic and Neolithic: evidence from stable isotope values of humans and dogs. Journal of Archaeological Science 34:2125-50.
Juliet Clutton-Brock, Nanna Noe-Nygaard. 1990. New osteological and C-isotope evidence on Mesolithic dogs: Companions to hunters and fishers at Star Carr, Seamer Carr and Kongemose. Journal of Archaeological Science, Volume 17, Issue 6, Pages 643-653.

Structure/figures/tables/raw data:
In Figure 2, there are several instances of one symbol that I do not understand nor find in the caption: A diamond overlain by the blue triangle, which is the symbol for turkey. Of course, this could happen accidentally in cases where e.g. a mammal has exactly the same isotope ratios as a turkey sample – but not so often, I assume. Please check.

In Figure 2 as well as in the caption of Figure 2, please indicate relative to which standard the d13C and d15N values are presented (e.g., “d15N (‰ AIR)”).

From the manuscript in its current form, I do not understand why it is necessary for your FRO analysis to calibrate the radiocarbon ages, e.g. using phase models or KDE plots in Figure 3. You write in line 157 that you use phase and KDE plots to graphically display the modelling results. However, there must be another reason for these calculations – please state it clearly either in the methods section or in the caption of Figure 3. As mentioned later, I think it is best to perform FRO calculations in 14C space, not calendar age space. However, I think it is useful, as you did in Table 3, to model your results with and without DeltaR on the dog bones to show the effect of the FROs.

In Tables 4 and 5, you could explain the abbreviations in the table caption (FRO, FMC, FDC).

Experimental design

Original primary research within Aims and Scope of the journal:
Yes/no comments.

Research question well defined, relevant & meaningful. It is stated how research fills an identified knowledge gap:
Yes/no comments.

Rigorous investigation performed to a high technical & ethical standard:
In the last part of the Results section, lines 198-218, you describe the connection between carbonate alkalinity and dead carbon fraction in fish. Specifically, you mention that a total alkalinity of >90mg/L CaCO3 in the water causes a dead carbon fraction of >0.05 in fish. However, I do not think that we should operate with threshold values like these; instead, increasing concentrations of 14C-free carbonate cause increasing FROs in fish, in a continuum. Later in the discussion, though, we can choose a threshold value to separate relevant from irrelevant FROs.

A minor correction: even though most of the total alkalinity in natural systems is carbonate alkalinity, you should consider to separate the two concepts in the text. The total alkalinity comprises all the bases in the solution, and not all of them contribute with carbon.

Methods described with sufficient detail & information to replicate:
The beginning of the sentence in lines 92-93 is a bit out of context: how have you established that fish accounted for 20-40 % of the dogs’ diet? By the archaeozoological analyses mentioned in the introduction, or by the isotope measurements that you mention later in the article?

Validity of the findings

Impact and novelty not assessed. Negative/inconclusive results accepted. Meaningful replication encouraged where rationale & benefit to literature is clearly stated.
No comments.

Data is robust, statistically sound, & controlled.
Yes – although I would not use calibrated radiocarbon ages to draw conclusions about reservoir offsets. Differences in radiocarbon ages, which are caused by carbon originating from different carbon reservoirs, are already visible on the radiocarbon timescale. When the ages are calibrated, an additional uncertainty is added to the ages from the shape and measurement uncertainty of the calibration curve. Therefore, small offsets are less visible on the calendar timescale than on the radiocarbon timescale.
In line 189, you state that you used the criteria by Van Klinken (1999) to assess the quality of the extracted collagen. However, as you use ultrafiltration (which Van Klinken did not use), you might want to consider using other criteria, e.g., lower collagen yields should be acceptable, as using a pretreatment method with ultrafiltration removes some of the sample mass.

In lines 162-165, you describe that you combined multiple dates with OxCal’s R_Combine function. I think it is better to use the Combine function in cases when the carbon derives from different samples. The R_Combine function should be used when the carbon derives from the same carbon reservoir, e.g. several samples from the same bone.

---

## Round 0.2 · accepted · Accept

All concerns of the reviewers were answered and the manuscript was improved. Please work with the production office to make the corrections suggested by the reviewer of your revised manuscript.

# ·

Basic reporting

all good, except for 2 text edits;
line 133 - insert "fish", i.e. "For example, fish isotope values were..."
line 170 - the square root sign should be outside the bracket

Experimental design

all good

Validity of the findings

all good

Additional comments

I have checked all the revisions and rebuttal comments and I am satisfied that the authors have addressed all my concerns